# Peer review of "Novel Single Inhibitor of HDAC6/8 and Dual Inhibitor of PI3K/HDAC6 as Potential Alternative Treatments for Prostate Cancer"

_pharmaceuticals, 2021, doi:10.3390/ph14050387_

Round 1

Reviewer 1 Report

The authors examined antitumor effects of a new HDAC6/8 inhibitor (LASSBio-1911) and a new dual-PI3K/HDAC6 inhibitor (LASSBio-2208) on human prostate cancer cell line PC3. In this study, they examined their effects on PC3 cells in terms of the cell viability, apoptosis, cleavage of caspase-3, cell cycle, p-JNK, p-STAT3, JAK2, IL-6 production and migration ability. The results suggest that the dual-PI3K/HDAC6 inhibitor seems to be a promising reagent and the mechanisms proposed are plausible. However, several points must be addressed and clarified.

Specific comments:

1) Figure 1. Why were the significant effects of reagents disappeared after 48 hr culture?  It seems strange. There is no explanation.

2) Figure 2. The flow cytometry figures (shape) are poor. The horizontal and vertical bars must be showed. The percentages of right lower (early apoptosis) in the Control or Vehicle are approximately 40%. However, in figure 2B, they are low. There is discrepancy.

3) Figure 3A. Why was cleavages Caspase-3 clearly detected in the Control and the Vehicle? Why was the cleavage caspase-3 decreased in higher doses of LASSBio-199?

4) Figure 4. Why was the expression of p-JNK in PC3 cells increased without any drug?

5) Can these drugs show any antitumor effects in xenograft mouse models?

Author Response

Comments and Suggestions for Authors

The authors examined antitumor effects of a new HDAC6/8 inhibitor (LASSBio-1911) and a new dual-PI3K/HDAC6 inhibitor (LASSBio-2208) on human prostate cancer cell line PC3. In this study, they examined their effects on PC3 cells in terms of the cell viability, apoptosis, cleavage of caspase-3, cell cycle, p-JNK, p-STAT3, JAK2, IL-6 production and migration ability. The results suggest that the dual-PI3K/HDAC6 inhibitor seems to be a promising reagent and the mechanisms proposed are plausible. However, several points must be addressed and clarified.

Specific comments:

1) Figure 1. Why were the significant effects of reagents disappeared after 48 hr culture?  It seems strange. There is no explanation.

Answer: The MTT assay is a colorimetric assay routinely used to determine cytotoxicity or drug safety. This is based on the formation of chromogen, selectively generated in viable cells. The principle of this method is the evaluation of the activity of mitochondrial dehydrogenases, quantified by the reduction of MTT (a water-soluble yellow salt) to formazan (purple crystals, insoluble in water) (LI & SONG, 2007). There are some published reports that the induction of cell death through the treatment of tumor cells with some drugs can play a crucial role in making the mitochondria overactive, leading to increased metabolic viability and reduced MTT. Therefore, the conclusions drawn about the growth inhibition induced by treatment with the compounds LASSBio 1911 and LASSBio 2208 based on metabolic viability assays, may not correlate with the inhibition of growth and/or loss of clonogenic survival, so we carry out others tests to confirm cell death from apoptosis, such as the apoptosis and effector caspase-3 activation assays. This result may also be related to the arrest of the cell cycle, which seems to be playing an important role as we can see in Table 1. As we already know, the pH changes in the medium and the increase in growth factors secreted by the tumor cell itself they may be interfering both directly in the test molecule, as well as in its binding to the receptor, according to the time that they are in the conditions of the cell culture.

References:

LI, J.; SONG, L. Applicability of the MTT assay for measuring viability of cyanobacteria and algae, specifically for Microcystis aeruginosa (Chroococcales, Cyanobacteria). Phycologia, v. 46, p. 593–599, 2007.

Rai, Y., Pathak, R., Kumari, N. et al. Mitochondrial biogenesis and metabolic hyperactivation limits the application of MTT assay in the estimation of radiation induced growth inhibition. Sci Rep 8, 1531 (2018).

Jabbar SA, Twentyman PR, Watson JV. The MTT assay underestimates the growth inhibitory effects of interferons. Br J Cancer. 1989 Oct;60(4): 523-8.

Please see discussion section, page 10, lines 261-264.

2) Figure 2. The flow cytometry figures (shape) are poor. The horizontal and vertical bars must be showed. The percentages of right lower (early apoptosis) in the Control or Vehicle are approximately 40%. However, in figure 2B, they are low. There is discrepancy.

Answer: Figure 2A was increased. Please see page 4. The images are representative of one assay. However each protocol was done in three different days. The graphs are the results of media/SD of all assays done in different days. It can be the reason for some diferences between an independet assay and the final graph.

3) Figure 3A. Why was cleavages Caspase-3 clearly detected in the Control and the Vehicle?

Answer: it may be that some cleavage caspase may be found in control cells due to any technical issue. Another possibility could be that it is the basal level for this cell type. Both explanations are hypothesis and we do not have another explanation.

Why was the cleavage caspase-3 decreased in higher doses of LASSBio-199?

Answer: This result may be related to the formation of nonspecific bonds in high concentrations, so that the compound binds to unwanted targets, activating or inhibiting accessory signaling pathways that can interfere with the studied effect. Or it may be due to the solubility of the compound and changes due to the pH of the solution. Also, there are aspects regarding these compounds being targets of other cellular enzymes and forming metabolites that can interfere with their activity.

4) Figure 4. Why was the expression of p-JNK in PC3 cells increased without any drug?

Answer: JNK-signaling pathway is required for growth of prostate carcinoma cells in vitro and in vivo. The JNK pathway is a target in the treatment of prostate carcinoma. JNK exerts functions to control a wide variety of cell processes including cell apoptosis, proliferation, migration, survival, differentiation and inflammation through activating a number of nuclear and non-nuclear molecules.

References:

Xu R, Hu J. The role of JNK in prostate cancer progression and therapeutic strategies. Biomed Pharmacother. 2020 Jan;121:109679.

Jiang W, Jiang C, Pei H, Wang L, Zhang J, Hu H, Lü J. In vivo molecular mediators of cancer growth suppression and apoptosis by selenium in mammary and prostate models: lack of involvement of gadd genes. Mol Cancer Ther. 2009 Mar;8(3):682-91.

Jiménez-Vacas JM, Herrero-Aguayo V, Gómez-Gómez E, León-González AJ, Sáez-Martínez P, Alors-Pérez E, Fuentes-Fayos AC, Martínez-López A, Sánchez-Sánchez R, González-Serrano T, López-Ruiz DJ, Requena-Tapia MJ, Castaño JP, Gahete MD, Luque RM. Spliceosome component SF3B1 as novel prognostic biomarker and therapeutic target for prostate cancer. Transl Res. 2019 Oct;212:89-103.

Please see item 2.4, page 6, lines 151-154.

5) Can these drugs show any antitumor effects in xenograft mouse models?

Answer:  There are already reports of the use of dual HDAC / tyrosine kinase inhibitors in xenographic mice where lung tumor and thyroid tumor were induced and the tested inhibitors showed promising results. There are still no reports of the use of these inhibitors in prostate tumors and despite being a different tumor, with different hormonal stimuli like those mentioned above, we hope that these inhibitors can also show promising results in models of xenographic mice.

References:

Zhang W, Zhang Y, Tu T, Schmull S, Han Y, Wang W, Li H. Dual inhibition of HDAC and tyrosine kinase signaling pathways with CUDC-907 attenuates TGFβ1 induced lung and tumor fibrosis. Cell Death Dis. 2020 Sep 17;11(9):765.

Kotian S, Zhang L, Boufraqech M, Gaskins K, Gara SK, Quezado M, Nilubol N, Kebebew E. Dual Inhibition of HDAC and Tyrosine Kinase Signaling Pathways with CUDC-907 Inhibits Thyroid Cancer Growth and Metastases. Clin Cancer Res. 2017 Sep 1;23(17):5044-5054.

Reviewer 2 Report

The authors characterized two compounds as PI3K/HDAC6 inhibitors in a prostate cancer cell line, PC3.  The authors investigated the effects of these two compounds on the proliferation, apoptosis, cell cycle, migration, IL-6 secretion, and JNK/JAK2/STAT3 activation in PC3 cells. This work provides new insight into the mechanism of action of these new compounds.  However, there are several major points need to be addressed before it can be considered for publication. 

  1. If they induce such levels of apoptosis (line 101), why they only showed temporary inhibition of cell viability (Figure 1)?
  2. All the experiments were done with only one cell line PC3. Validation in additional cell lines (at lease 1~2 more cell lines) will strengthen the manuscript.
  3. What is the rationale of checking JNK pathway? This is not justified or described in the manuscript. 
  4. In all the experiments, several different concentrations of compounds were used. However, we did not observed any dose-dependent responses, except in the migration assay.
  5. Discussion part is too long. Some results were repeated in the discussion and should be deleted. Some parts should be moved to the introduction as background. In introduction, since PI3K and HDAC are not the main focus of this study, their introduction can be more brief. 

Author Response

Comments and Suggestions for Authors

The authors characterized two compounds as PI3K/HDAC6 inhibitors in a prostate cancer cell line, PC3.  The authors investigated the effects of these two compounds on the proliferation, apoptosis, cell cycle, migration, IL-6 secretion, and JNK/JAK2/STAT3 activation in PC3 cells. This work provides new insight into the mechanism of action of these new compounds.

However, there are several major points need to be addressed before it can be considered for publication. 

1) If they induce such levels of apoptosis (line 101), why they only showed temporary inhibition of cell viability (Figure 1)?

Answer: The MTT assay is a colorimetric assay routinely used to determine cytotoxicity or drug safety. This is based on the formation of chromogen, selectively generated in viable cells. The principle of this method is the evaluation of the activity of mitochondrial dehydrogenases, quantified by the reduction of MTT (a water-soluble yellow salt) to formazan (purple crystals, insoluble in water) (LI & SONG, 2007). There are some published reports that the induction of cell death through the treatment of tumor cells with some drugs can play a crucial role in making the mitochondria overactive, leading to increased metabolic viability and reduced MTT. Therefore, the conclusions drawn about the growth inhibition induced by treatment with the compounds LASSBio 1911 and LASSBio 2208 based on metabolic viability assays, may not correlate with the inhibition of growth and/or loss of clonogenic survival, so we carry out others tests to confirm cell death from apoptosis, such as the apoptosis and effector caspase-3 activation assays. This result may also be related to the arrest of the cell cycle, which seems to be playing an important role as we can see in Table 1. As we already know, the pH changes in the medium and the increase in growth factors secreted by the tumor cell itself they may be interfering both directly in the test molecule, as well as in its binding to the receptor, according to the time that they are in the conditions of the cell culture.

2) All the experiments were done with only one cell line PC3. Validation in additional cell lines (at lease 1~2 more cell lines) will strengthen the manuscript.

Answer: We have already published a previous paper using DU145 and RWPE-1 cell lines, both prostate cancer cell line, in the face of treatment with these inhibitors and we obtained promising results evaluating parameters such as cytotoxicity and cell migration.

Reference:

Rodrigues, D.A.; Guerra, F.S.; Sagrillo, F.S.; de Sena, M.; Pinheiro, P.; Alves, M.A.; Thota, S.; Chaves, L.S.; Sant'Anna, C.M.R.; Fernandes, P.D.; Fraga, C.A.M. Design, synthesis, and pharmacological evaluation of first-in-class multitarget n-acylhydrazone derivatives as selective HDAC6/8 and PI3Kα inhibitors, ChemMedChem 2020, 15, 539-551.

Please see discussion section, page 264-269

3) What is the rationale of checking JNK pathway? This is not justified or described in the manuscript. 

Answer: JNK-signaling pathway is required for growth of prostate carcinoma cells in vitro and in vivo. The JNK pathway is a target in the treatment of prostate carcinoma. JNK exerts functions to control a wide variety of cell processes including cell apoptosis, proliferation, migration, survival, differentiation and inflammation through activating a number of nuclear and non-nuclear molecules.

References:

Xu R, Hu J. The role of JNK in prostate cancer progression and therapeutic strategies. Biomed Pharmacother. 2020 Jan;121:109679.

Jiang W, Jiang C, Pei H, Wang L, Zhang J, Hu H, Lü J. In vivo molecular mediators of cancer growth suppression and apoptosis by selenium in mammary and prostate models: lack of involvement of gadd genes. Mol Cancer Ther. 2009 Mar;8(3):682-91.

Jiménez-Vacas JM, Herrero-Aguayo V, Gómez-Gómez E, León-González AJ, Sáez-Martínez P, Alors-Pérez E, Fuentes-Fayos AC, Martínez-López A, Sánchez-Sánchez R, González-Serrano T, López-Ruiz DJ, Requena-Tapia MJ, Castaño JP, Gahete MD, Luque RM. Spliceosome component SF3B1 as novel prognostic biomarker and therapeutic target for prostate cancer. Transl Res. 2019 Oct;212:89-103.

Please see page 6, lines 151-154.

4) In all the experiments, several different concentrations of compounds were used. However, we did not observed any dose-dependent responses, except in the migration assay.

Answer: In biological studies the response to concentration can be complex and is often non-linear. This result may be related to the formation of nonspecific bonds in high concentrations, so that the compound binds to unwanted targets, activating or inhibiting accessory signaling pathways that can interfere with the studied effect. Or it may be due to the solubility of the compound and changes due to the pH of the solution. Also, there are aspects regarding these compounds being targets of other cellular enzymes and forming metabolites that can interfere with their activity.

5) Discussion part is too long. Some results were repeated in the discussion and should be deleted. Some parts should be moved to the introduction as background. In introduction, since PI3K and HDAC are not the main focus of this study, their introduction can be more brief. 

Answer: we changed some parts of introduction and discussion to improve the text. Please, see:

Introduction section, page 2, lines 33-60

Discussion section:

1st paragraph, Lines 248-250; 261-269

2nd paragraph, lines 270-277

4th paragraph, lines 286-296

5th paragraph, lines 300-307

6th paragraph, lines 312-322

Round 2

Reviewer 1 Report

The authors respond to my comments properly. The study could provide useful information for the readers of Pharmaceuticals.